# ^1^H NMR-Based Metabolomics Reveals the Intrinsic Interaction of Age, Plasma Signature Metabolites, and Nutrient Intake in the Longevity Population in Guangxi, China

**DOI:** 10.3390/nu14122539

**Published:** 2022-06-18

**Authors:** He Li, Minhong Ren, Quanyang Li

**Affiliations:** College of Light Industry and Food Engineering, Guangxi University, Nanning 530004, China; 1816401001@st.gxu.edu.cn (H.L.); 1716402006@st.gxu.edu.cn (M.R.)

**Keywords:** nutrient intake, longevity, centenarians, metabolomic profiling, ^1^H NMR, healthy aging

## Abstract

Health and longevity populations have distinct metabolic and nutrient intake profiles. However, the relationship between biomarkers of longevity-related metabolites and dietary nutrient intake profiles, as well as metabolic markers associated with longevity features, have not been fully elucidated. Therefore, ^1^H nuclear magnetic resonance (^1^H NMR)-based plasma metabolomics profiling was conducted in the present study to identify potential metabolites which can be used as specific markers for the evaluation of healthy aging. Plasma samples were obtained from centenarians and nonagenarians from the longevous region, and elderly participants aged 60–89 from the longevous region, as well as a low centenarian ratio region. The results showed that participants from longevous regions exhibited higher plasma levels of citrate, tyrosine, choline, carnitine, and valine, as well as lower contents of VLDL, lactate, alanine, *N*-acetyl glycoprotein (NAG), trimethylamine oxide (TMAO), α-glucose, β-glucose, and unsaturated lipids. The differential plasma metabolites were associated with an alteration in glycolysis/gluconeogenesis; aminoacyl-tRNA biosynthesis; alanine, aspartate, and glutamate metabolism; and phenylalanine, tyrosine, and tryptophan biosynthesis in participants from longevous regions. The signature metabolites were associated with higher dietary fiber intake, as well as lower energy and fat intake. The results of the present study demonstrate key longevity signature metabolites in plasma, and the dietary patterns identified provide a basis for further health and longevity research.

## 1. Introduction

Health and longevity have become a key focus of the scientific and social community owing to the rapid increase in the global aging population. Diet is an essential factor in sustaining human life and maintaining healthy longevity, and dietary changes result in different effects on the organism [1]. For example, calorie restriction prolongs life span, and abrogates the occurrence of age-related diseases and metabolic disorders [2]. Notably, a Mediterranean diet reduces incidence of cardiovascular disease [3]. Previous findings indicate that a nutritionally unbalanced diet, such as a high-fat and high-calorie diet, promotes the occurrence of metabolic disorders and contributes to the onset of several chronic diseases, ultimately accelerating the aging process [4]. Furthermore, the changes in dietary patterns that results in the metabolite variations have also been demonstrated. A previous study reported that adherence to a Mediterranean diet is significantly and positively correlated with plasma levels of citric acid and pyruvate metabolites [5].

Metabolomics is a rapidly developing field of science that entails the study of the relationship between various diseases and the metabolic profile of cells, tissue, and biofluids. Metabolite profiles are determined using techniques such as ^1^H nuclear magnetic resonance (^1^H NMR) spectroscopy and mass spectrometry. Untargeted metabolomics enables a comprehensive evaluation of diverse metabolite alterations, which reflect changes in metabolic pathways caused by modulation of the biological system [6]. Numerous mechanisms have been implicated in the aging process, including oxidative stress, lipid peroxidation, and the inflammatory response related to metabolic alterations [7,8]. ^1^H NMR spectroscopy has high clinical significance, since metabolomic profiling can be easily conducted using peripheral tissue, plasma, fecal, and urine samples. Numerous studies have explored the profile of metabolites in serum, plasma, saliva, and urine and their association with diabetes mellitus, cardiovascular disease, Alzheimer’s disease, and cancer [9,10,11]. Metabolite profiles are associated with health and biological aging in humans [12]. Amino acid and lipid metabolites are associated with the health of humans and the process of biological aging. Metabolites of the TCA cycle, including citrate, and metabolites from phenylalanine metabolism are correlated with biological age [13]. Populations with long lifespans, especially centenarians, are likely to exhibit healthy and beneficial aging patterns. The metabolic profile of long-lived people may have a key relationship with the diet they consume. The specific mechanisms of the aging process have not been fully elucidated. Notably, evaluating the metabolic differences that exist in the long-lived elderly can play an important role in elucidating the mechanisms underlying healthy aging. Therefore, it is imperative to explore the characteristics of dietary and metabolite profiles in long-lived populations [13]. Currently, only a few studies have explored the relationship between signature plasma metabolites and nutrient intakes in healthy long-living people. Metabolomics is an effective approach for nutritional assessment as it is used for identification of various small molecules present in biological systems, and untargeted metabolomics can offer a clear understanding of how humans respond to complex diets [14]. ^1^H NMR spectroscopy, combined with multivariate statistical analysis, can be used for the identification of characteristic metabolites associated with healthy longevity in different biofluid samples.

The Hongshui River Basin is located in the Hechi region of Guangxi province, China, and is a world-renowned longevity area. This region is characterized by a high proportion of healthy and long-lived elderly people, a low genetic diversity background, as well as long-term stable dietary habits and lifestyles. These features make the region an ideal study area for exploring the mechanisms underlying healthy longevity. Findings from our previous study showed that the fecal metabolites of healthy, long-lived elderly people in the Hechi region of Guangxi province have high levels of short-chain fatty acids, which were positively correlated with dietary fiber intake. The findings showed that specific metabolic patterns and dietary characteristics are correlated with health and longevity in humans [15]. In addition, an examination of physical health indicators in subjects of the longevity regions of Hechi by our research team revealed that long-lived older adults, especially centenarians, had a good hepatic function, lower lipid levels, and lower levels of inflammation. This makes it more meaningful to explore the health and longevity phenomenon in Guangxi from the perspective of metabolomics in combination with dietary characteristics [16]. In the present study, diet-related signature metabolites of longevity were evaluated through the analysis of plasma metabolites and dietary characteristics of the longevity population in the Hechi region. The findings of the current study have the potential to provide a theoretical basis for establishing healthy dietary plans, exploring mechanisms underlying the aging process, and designing strategies for achieving healthy aging.

## 2. Materials and Methods

### 2.1. Enrollment of Participants

The study was conducted in Donglan, Fengshan, and Dahua counties, which are located in the Hongshui River Basin; the counties are part of the Hechi region, Xixiangtang District, Nanning City, Guangxi province, China. The centenarian incidence of the four counties was 31.45, 28.49, 14.97, and 0.26 centenarians/100,000 citizens, respectively, in the year that sampling was conducted. The Xixiangtang District had a low centenarian ratio and was selected as the control group (NLR). Healthy centenarians and nonagenarians in the longevous region (LR), and elderly people aged 60–89 in the LR and the NLR were recruited to the study according the population information provided by local government authorities and information obtained through household visits. Information on the age, physical condition, medical history, and food habits of the participants was obtained through a survey. Anthropometric data (including weight, height, and BMI) were obtained during the household visits. Ages of all individuals were recorded and verified. Participants who had undergone medical treatment or antibiotic therapy six months prior to the plasma sample collection were excluded from the study. A total of 90 people from the LR group and 27 people from the NLR group were enrolled in this study. The study comprised 27 centenarians in the LRC group (age: 100–118 years), 37 nonagenarians in the LRN group (age: 90–99 years), 26 elderly people in the LRE group (age: 60–89 years), and 27 elderly people in the NLRE group (age: 60–89 years) (Table 1 and Appendix A).

The Ethics Committee of Guangxi University approved the present study (Approval No.: GXU-M-2019003). The study was conducted following the guidelines of the Declaration of Helsinki. All participants provided written informed consent before participation in the study.

### 2.2. Assessment of Dietary Nutrition Status

The dietary nutritional status of participants was assessed using a semi-quantitative food frequency questionnaire (FFQ) and consecutive 7-day weighed dietary records [17,18]. All participants were requested to dine alone throughout the nutritional survey and were urged to maintain their usual eating patterns. Trained investigators weighed and documented every food and beverage consumed by the subjects using electronic food scales, measuring cups, and spoons. Standard serving sizes were utilized for unquantifiable foods. Qualified dietitians evaluated and verified the dietary data at a dietary survey location for quality control purposes. Dietary data was converted to nutritional consumption levels using Chinese food composition tables. The average daily nutritional intake was determined by multiplying the food consumed (in grams) or the portion size consumed by the nutrient content per 100 g of the food listed in the Chinese food composition tables [19,20].

### 2.3. Sample Collection and Preparation

The study subjects fasted for 12 h before plasma samples were collected to minimize food interference. Blood samples were transferred into vacuum tubes containing heparin anticoagulant and were gently shaken. The samples were stored for 1 h at room temperature, then centrifuged at 3000 r/min for 10 min. Plasma supernatant samples were collected, transported to the laboratory, then immediately stored at −80 °C for subsequent analysis.

The plasma samples were thawed on ice and centrifuged at 12,000 r/min for 10 min at 4 °C to remove debris. Further, 200 µL of the plasma sample was mixed with 400 µL of sodium phosphate buffer (45 mM K2HPO4/NaH2PO4, 0.9% NaCl, and 15% D2O, pH = 7.4) in 2 mL microcentrifuge tubes, then centrifuged at 12,000 r/min for 20 min at 4 °C. A total of 550 µL of the supernatant was transferred into a 5 mm NMR tube.

### 2.4. NMR Data Acquisition and Analysis

The NMR spectra of all samples were acquired using a Bruker AVANCE 500 MHz NMR Spectrometer (Bruker Biospin, Rheinstetten, Germany). A pre-saturated Carr–Purcell–Meiboom–Gill (CPMG) pulse sequence, with the following parameters: number of samples (NS) = 64, temperature = 25 °C, spectral width (SWH) = 10,000 Hz, determination frequency (SF) = 500.13 MHz, relaxation delay (D1) = 2 s, with number of data points (TD) = 65,536, sampling time (AQ) = 3.277 s, and fixed echo time (D20) = 2 ms, number of cycles (L4) = 20, (O1P) = 4.7 ppm, FID resolution = 0.153 was used for spectrometric analysis. ^1^H NMR spectra were baseline corrected and phase adjusted. Subsequently, the chemical shift was calibrated, interference of water peak was removed, then divided into buckets and normalized using MestReNova software (14.0.0-23239-win, Mestrelab Research, Santiago, Spain). The proton signal of α-glucose (δ = 5.233 ppm) was utilized to calibrate the chemical shift to eliminate systematic inaccuracy in the chemical shift. The spectra were then separated into a 0.001 ppm wide segments range from δ 0.50 to δ 9.00 ppm. Notably, the regions at δ 4.200–δ 5.100 ppm were excluded as the water peak. All spectra variables were normalized to the total surface area under the peak curves for effective comparisons between spectra of different samples. 

### 2.5. Plasma Metabolites Identification

The plasma metabolites were identified by comparing the NMR spectra with the chemical shifts and ^1^H spectral patterns of reference compounds from the Chenomx spectral database library (Chenomx software NMR suite 8.1, Chenomx, Inc., Alberta, AB, Canada) of small molecule metabolites for 500 MHz (11.7 Tesla) magnetic field strength NMR. These metabolites were then compared with metabolites in the HMDB (http://www.hmdb.ca/, accessed on 3 January 2022) and BMRB (http://www.bmrb.wisc.edu/, accessed on 17 January 2022) NMR spectral databases and findings from previous studies [9,21].

### 2.6. Statistical Analysis

The results were presented as mean values ±standard deviation. The nonparametric Kruskal–Wallis test, conducted using SPSS 22.0 software (International Business Machines Corp., Armonk, NY, USA), was used for analysis of the statistical differences between the four groups. NMR spectra data were imported into SIMCA-P 13.0 software (Umetrics, Umeå, Sweden) for multivariate analysis. Principal component analysis (PCA), partial least squares-discriminant analysis (PLS-DA), and orthogonal projections to latent structures discriminant analysis (OPLS-DA) were performed to identify the differential metabolites among participants from different regions. Additionally, permutation testing (*n* = 200) was conducted to assess the validity of the model used to evaluate differential metabolites. Metabolites with variable importance in projection (VIP) values > 1 and (|*p* (corr)| ≥ 0.5) from the OPLS-DA model were considered as differential metabolites. Integral values of the differential metabolites between the two groups were then statistically analyzed using the Mann–Whitney U test in SPSS 22.0 software. *p* < 0.05 represented a significant difference. Metaboanalyst 5.0 tool (https://www.metaboanalyst.ca, accessed on 4 March 2022) was used for pathway enrichment analysis to identify metabolic pathways linked with longevity associated with the differential metabolites.

A Spearman correlation test was conducted to explore correlations between the nutrient intake and differential plasma metabolites (SPSS 22.0, International Business Machines Corp., Armonk, NY, USA). The analysis was adjusted by age, gender, and BMI. Pearson’s analysis was performed to determine correlations between age and levels of longevity-related plasma signature metabolites in participants from the longevous region, after adjustment for gender and BMI. Graphs were generated using GraphPad Prism 8 (GraphPad Software, La Jolla, CA, USA).

## 3. Results

### 3.1. Nutrient Intakes in Participants

The nutrient intake levels in the four groups are presented in Table 2. The results showed that nutrient intakes were significantly different among populations from the different regions. Notably, the level of energy, protein, fat, saturated fatty acid (SFA), monounsaturated fatty acids (MUFA), polyunsaturated fatty acids (PUFA), cholesterol, folic acid, nicotinic acid, and sodium were significantly lower in subjects in the LRC, LRN, and LRE groups compared with the levels in participants in the NLRE group (*p* < 0.01). Levels of energy, protein, fat, cholesterol, and sodium intake were lower by 24.63%, 34.60%, 47.80%, 49.81%, and 28.28%, respectively, for the LRC group; were lower by 19.73%, 32.85%, 44.75%, 47.75%, and 26.54%, respectively, for the LRN group; and were lower by 8.98%, 10.54%, 18.14%, 17.60%, and 21.63%, respectively, for the LRE group compared with the levels for the NLRE group. Furthermore, the levels of intake of carbohydrate, dietary fiber, vitamin A, choline, magnesium, and copper intake were significantly higher in the LRC, LRN, and LRE groups relative to the levels in the NLRE group (*p* < 0.01). The results of this study were consistent with findings from a previous dietary survey conducted by our research group in Bama County, which is another longevity area in the Guangxi region [15]. These findings indicate that lower energy, fat, cholesterol intake, and higher dietary fiber and vitamin A intake are positively correlated with good health and longevity. In addition, the results showed that the intake of choline, magnesium, and copper improves the health of the elderly.

### 3.2. ^1^H NMR Blood Plasma Metabolites Identification

^1^H NMR spectroscopy was conducted to identify differential metabolites among participants from different regions. Analysis of the NMR spectra for the four groups revealed a consistent spectrum of metabolite signals. A total of 35 metabolites were identified from the NMR spectra. The identified plasma metabolites for each aging group are presented in Figure 1. Assignments of different metabolites, with their chemical shifts, are shown in Appendix A.

### 3.3. Multivariate Analysis of the Plasma Metabolites

PCA was performed to explore the overall distribution of plasma samples between groups. PLS-DA was further performed to explore differences in plasma metabolites of the different groups. Moreover, permutation tests (*n* = 200 replicates) were conducted to verify the validity of the model, and OPLS-DA was performed for the identification of differential metabolites. PCA and PLS-DA results for the different groups are shown in Appendix A. PCA, PLS-DA, and OPLS-DA distinctively separated participants from the different regions (Figure 2). The value of the PCA model parameters R2X (cum) in the LRC and NLRE, LRN and NLRE, and LRE and NLRE groups were 0.826, 0.843, and 0.848, respectively. Values more than 0.4 indicate that the PCA models are reliable and effective in distinguishing the various groups. The PLS-DA model showed a clear distinction between LRC and NLRE, LRN and NLRE, and LRE and NLRE groups, with R2Y (cum) and Q2 (cum) values of (0.947, 0.909), (0.936, 0.915), and (0.946, 0.913), respectively. The PLS-DA model was further validated through permutation testing. The findings of the permutation test showed that LRC and NLRE, LRN and NLRE, and LRE and NLRE were within the R2Y intercept < 0.4 and Q2Y intercept < 0.05, and the leftmost data of the R2 and Q2 were less compared with the rightmost values, indicating that the model was valid and reliable. OPLS-DA results showed that the R2Y (cum) and Q2 (cum) values between LRC and NLRE, LRN, and NLRE, and LRE and NLRE, were (0.947, 0.916) (Figure 3A), (0.936, 0.915) (Figure 3C), and (0.971, 0.928) (Figure 3E). R2Y and Q2 are key parameters in evaluating the quality of OPLS-DA models. The prediction rate Q2 > 0.4 indicates that the model is reliable. Therefore, the quality parameters of the OPLS-DA model indicated that the model effectively explained the differences in plasma metabolites between the two groups and was accurate in the identification of metabolites that were differential between groups. The variable importance in projection (VIP) values > 1, combined with the s-plot plots of each group (Figure 3B,D,F) and the (|*p* (corr)| ≥ 0.5) values of the OPLS-DA model, indicated differential metabolites among the groups.

### 3.4. Differential Analysis of Potential Plasma Metabolic Markers Associated with Longevity

Differential metabolites in plasma samples from the LRC, LRN, LRE, and NLRE groups were identified using the OPLS-DA model. The model was used to explore the effect of differential metabolites levels on the health of participants from longevity regions and the control regions. The logarithmic values (base 2) of relative peak area ratios of the differential metabolites levels in the LRC, LRN, and LRE groups were determined and compared with those of the NLRE group. Further, the fold change (FC) between the groups was obtained (Table 3). Lipid (mainly VLDL), lactate, alanine, NAG, TMAO, and α-glucose contents in the LRC, LRN, and LRE groups were significantly lower relative to the contents in the NLRE group (*p* < 0.01). The results showed that citric acid, tyrosine, valine, choline, and carnitine contents were higher in the LRC, LRN, and LRE groups compared with the contents in the NLRE group (*p* < 0.01). In addition, plasma metabolites of the LRN and LRE groups had lower levels of β-glucose (*p* < 0.01) compared with the levels in the NLRE group. Moreover, the levels of unsaturated lipids in the LRC and LRN groups was significantly lower relative to the level in the NLRE group (*p* < 0.01). A box plot of the differential metabolites levels among the four groups is presented in Figure 4.

### 3.5. Enrichment Analysis of Metabolic Pathways Associated with Differential Metabolites

The MetaboAnalyst 5.0 tool was used to conduct enrichment analysis to explore the metabolic pathways associated with longevity. A total of 23 pathways were implicated in central metabolism, including glycolysis/gluconeogenesis; aminoacyl-tRNA biosynthesis; alanine, aspartate and glutamate metabolism; phenylalanine, tyrosine and tryptophan biosynthesis; valine, leucine, and isoleucine biosynthesis; ubiquinone and other terpenoid-quinone biosynthesis; phenylalanine metabolism; glycerolipid metabolism; pantothenate and CoA biosynthesis; the citrate cycle (TCA cycle); fructose and mannose metabolism; selenocompound metabolism; pyruvate metabolism; lysine degradation; galactose metabolism; glyoxylate and dicarboxylate metabolism; glycine, serine, and threonine metabolism; glycerophospholipid metabolism; amino sugar and nucleotide sugar metabolism; fatty acid degradation; valine, leucine and isoleucine degradation; *N*-glycan biosynthesis; and tyrosine metabolism (Figure 5 and Appendix A). The findings showed that four metabolic pathways were significantly enriched (*p* < 0.05). These four metabolic pathways included glycolysis/gluconeogenesis (*p* < 0.01); aminoacyl-tRNA biosynthesis (*p* < 0.01); and alanine, aspartate and glutamate metabolism (*p* < 0.05); and phenylalanine, as well as tyrosine and tryptophan, biosynthesis (*p* < 0.05).

### 3.6. Correlation Analysis between Nutrient Intake and Plasma Metabolites

A Spearman correlation analysis was conducted to explore the relationships between nutrient intake and differential metabolites levels. The results were then presented as a correlation heatmap (Figure 6). The level of valine was significantly positively correlated with energy intake, as well as the amounts of protein, fat, SFA, MUFA, PUFA, carbohydrate, dietary fiber, copper, and manganese consumed by the participants (*p* < 0.05). Choline levels were significantly positively correlated with energy intake, as well as consumed fat, SFA, MUFA, PUFA, carbohydrate, dietary fiber, choline, copper, and manganese levels (*p* < 0.05). The results showed that choline levels were significantly negatively correlated with the intake of vitamin B6, nicotinic acid, vitamin K, potassium, and magnesium (*p* < 0.05). Tyrosine levels were significantly positively correlated with energy intake and amounts of protein, fat, SFA, MUFA, cholesterol, dietary fiber, nicotinic acid, and selenium consumed by the participants (*p* < 0.05). Citrate levels were significantly negatively correlated with folic acid and potassium intake (*p* < 0.05). The level of lactate was significantly positively correlated with the intake of riboflavin, vitamin B6, folic acid, potassium, and magnesium (*p* < 0.05). Lactate levels were significantly negatively correlated with the intake of energy, SFA, PUFA, carbohydrate, dietary fiber, choline, and manganese (*p* < 0.05). NAG levels were significantly positively correlated with the intake of cholesterol, folic acid, vitamin K, and magnesium (*p* < 0.05), and significantly negatively correlated with the intake of energy, SFA, MUFA, PUFA, carbohydrate, dietary fiber, copper, and manganese (*p* < 0.05). VLDL levels were significantly positively correlated with the intake of riboflavin, folic acid, potassium, magnesium (*p* < 0.01), and calcium (*p* < 0.05). On the contrary, VLDL levels were significantly negatively correlated with the intake of energy, SFA, MUFA, PUFA, dietary fiber, copper, and manganese (*p* < 0.05). Levels of unsaturated lipids were significantly positively correlated with folic acid intake (*p* < 0.05), and significantly negatively associated with energy intake, and the amounts of SFA, MUFA, dietary fiber, nicotinic acid, copper, and manganese consumed (*p* < 0.05). Alanine levels were significantly negatively associated with energy intake, as well as the amounts of saturated fatty acid, MUFA, carbohydrates, dietary fiber, and choline consumed (*p* < 0.05). Carnitine levels were significantly positively associated with the intake of vitamin B1, riboflavin, vitamin B6, and potassium (*p* < 0.05), and significantly negatively correlated with the intake of vitamin A, vitamin C, calcium, and manganese (*p* < 0.05). Notably, the correlation coefficients between dietary fiber intake and choline (r = 0.431, *p* < 0.01), NAG (r = −0.412, *p* < 0.01), and lactate (r = −0.401, *p* < 0.01) metabolites were high.

### 3.7. Change in Plasma Signature Metabolites Is Associated with Age in Longevous Region Participants

A Pearson correlation analysis was conducted to evaluate the relationships between the longevity-related plasma signature metabolites and the age of participants in longevous regions. The results showed that the levels of unsaturated lipids were significantly negatively correlated with age, whereas the levels of α-glucose, β-glucose, and TMAO were significantly positively correlated with age, although with a very low intensity (Figure 7). Notably, the levels of VLDL, lactate, alanine, NAG, citrate, tyrosine, choline, carnitine, and valine were not significantly correlated with age (Figure 7).

## 4. Discussion

^1^H NMR-based metabolomics is a widely used approach that does not require sample preparation, chromatography, or analyte ionization, enabling the effective identification of analytes, making it an ideal method for cohort studies [22]. In the present study, the profiles of metabolites in the plasma of healthy centenarians, nonagenarians, and the elderly from the LR were evaluated using the ^1^H NMR-based metabolomic method. Thirteen plasma metabolites, namely VLDL, lactate, alanine, NAG, citrate, tyrosine, choline, carnitine, TMAO, β-glucose, α-glucose, valine, and unsaturated lipids were identified in participants from the longevous region based on the OPLS-DA model and quantified by fold change criterion. The findings showed that unsaturated lipids, α-glucose, β-glucose, and TMAO were significantly correlated with the age of longevous region participants. Enrichment analysis showed that glycolysis/gluconeogenesis; aminoacyl-tRNA biosynthesis; alanine, aspartate and glutamate metabolism; and phenylalanine, tyrosine, and tryptophan biosynthesis pathways were significantly enriched in LR participants and may be the mechanisms underlying the relationship between key metabolites and age. Moreover, correlation analysis indicated significant associations between nutrient intake and plasma metabolites. The findings showed distinctive features of plasma metabolites in long-lived people from the LR, which are attributed to nutrient intake.

Elevated plasma lipid and unsaturated fatty acids levels increase the risk of age-related diseases, including cancers, diabetes, cardiovascular disease, and metabolic diseases [13]. The metabolic levels of VLDL in the LRC, LRN, and LRE, and the metabolic levels of unsaturated lipids in the LRC and LRN, were lower compared with the levels of these metabolites in the NLRE. Notably, the levels of unsaturated lipids were significantly negatively correlated with age in people from the LR. Additionally, analysis of nutrient intake showed that the three groups in the LR had significantly lower fat, SFA, MUFA, PUFA, and cholesterol intake relative to the intake among participants in the NLRE group. These results indicate that lower levels of lipid metabolism are perhaps a key characteristic of long-lived individuals. Furthermore, the findings of the present study revealed higher levels of α-glucose, β-glucose, and TMAO in centenarians than in other age groups, which indicated that the healthy aging process in long-lived populations may be accompanied by a decrease in efficiency of plasma glucose and TMAO metabolism [23].

The upregulation of gluconeogenesis and the inhibition of the TCA cycle are principal metabolic features of diabetes and related metabolic syndromes [24]. Increased oxidative stress induces excessive glycogen depletion and the production of large amounts of lactate during the aging of organisms. Elevated lactate level is associated with high mortality [25]. In the current study, the levels of lactate and α-glucose were lower in the blood of participants in the LR, and lower levels of β-glucose were observed in the LRN and LRE residents compared with the levels in the NLRE residents. The results showed that dietary fiber and manganese intake were negatively associated with lactate level. This implies that a suitable increase in dietary fiber and micronutrients in the daily diet is an effective approach for maintaining a healthy energy metabolism.

Citrate is an essential intermediate product in the TCA cycle, which is also involved in amino acid and lipid metabolism. The dysregulation of citrate metabolism is associated with the molecular pathophysiology of illnesses such as hypertension, atherosclerosis, and diabetes. A previous study reported that the serum citrate content was significantly higher in centenarians compared with the level in older adults with a mean age of 70 ± 6 years [13]. Retaining citrate in the mitochondria to sustain the TCA cycle is associated with longevity and elevated plasma citrate levels. Previous findings indicate that increased plasma citrate levels inhibit glycolysis [26]. The findings in the present study showed that participants from the LR had higher plasma citrate levels compared with the levels in participants from NLRE. This implies that participants from the LR have a significantly enriched energy metabolism relative to that of participants from the NLRE. Further analysis showed that participants in the LR had a higher intake of fruits, vegetables, and coarse grains compared with that of participants in the NLRE group. Notably, these foods are positively correlated with citrate levels [5]. This may be the reason underlying the higher plasma citrate contents in participants among different age groups in the LR compared with the levels in the NLRE group.

Alanine and tyrosine are non-essential amino acids, and valine is a glucogenic BCAA. Higher blood alanine levels are associated with insulin resistance and diabetes mellitus in older adults [10]. Levels of tyrosine and valine are lower in the plasma of cancer patients relative to the levels in healthy controls [9]. In addition, a previous study reported lower blood concentrations of valine in subjects with Alzheimer’s disease compared with the levels in healthy controls [11]. In the present study, the levels of tyrosine and valine were higher in participants from the LR, whereas the levels of alanine were lower compared with the levels in NLRE residents. The findings are consistent with results from a study conducted by Ma et al. [27]. High levels of NAG activate the inflammatory response. Previous studies on centenarians reported high circulating levels of proinflammatory molecules, and a high level of NAG [28]. On the contrary, the levels of NAG in the centenarians from the LR were significantly lower relative to the levels in the NLRE residents in the present study. A strong positive correlation between NAG levels and high energy intake has been reported in previous studies. The findings of the present study showed a significant negative association between dietary fiber intake and NAG levels in long-lived participants, indicating that inflammatory responses were inhibited in response to a dietary pattern with lower energy intake and higher dietary fiber intake among participants from the longevous regions.

Choline and its derivatives are required for normal functioning of the liver, muscles, and brain, and are primary constituents of cell membranes [29]. A study comprising a community-based cohort of 1391 nondemented subjects (age range: 36–83 years) reported that high choline intake was associated with better cognitive function [30]. In the present study, the findings indicated that dietary choline intake and plasma choline levels were significantly higher in elderly people, of all age groups in the LR, relative to the levels of subjects from NLRE. In addition, nutrient intake of dietary fiber, choline, copper and manganese were positively correlated with the plasma levels of choline metabolite. Carnitine plays a role as a shuttle for transporting fatty acids from the cytosol to the mitochondrial matrix, where β-oxidation occurs [31]. In the current study, high plasma carnitine content was observed in LR participants, which indicates enhanced β-oxidation in these subjects. A previous study reported a correlation between reduced plasma carnitine and human frailty, which is consistent with the results of the current study [32]. Notably, carnitine supplementation exhibits neuroprotective effects [33]. The relationship between choline and carnitine—TMAO—has not been fully elucidated. Microbes convert choline, betaine, and carnitine to trimethylamine (TMA), which is then oxidized to TMAO by the insulin-regulated enzyme flavin-containing monooxygenase 3 in the host liver. TMAO is closely associated with the occurrence of cardiovascular and several related diseases. Previous studies report that the intake of fish, seafood, and foods lacking dietary fiber result in higher contents of plasma TMAO [34,35]. In the present study, a lower plasma level of TMAO was observed in participants from the LR relative to the levels in subjects from the NLRE. The findings indicate that participants from the LR had enriched metabolic processes that downregulated plasma TMAO levels. This implies that the distinct nutritional features in residents of the LR promoted health and longevity of the subjects.

Multivariate analysis showed insignificant variance in plasma metabolites between the three groups in the LR. This can be attributed to the identical dietary habits, nutrition intake, and lifestyles of senior individuals sampled from the LR. In addition, the subtle differences can be ascribed to the similar geographical environment in the LR. This implies that geographical environment has a significant impact on the metabolites of healthy elderly individuals. Diet, lifestyle, and other environmental variables markedly affect metabolite levels, resulting in regional metabolomic phenotypes [36,37]. The Hechi region, located in Guangxi province, is a well-known longevity region in China. The findings from the dietary survey showed that the local long-lived elderly prefer a high-fiber diet with corn, sweet potato, and rice porridge as a staple foods, complemented with several dark vegetables, such as pumpkin, pumpkin seedlings, and sweet potato leaves. Notably, the residents of this region rarely consumed dairy products and seafood. The high prevalence of centenarians, consistent lifestyle, and low immigration rates make the region an ideal geographic area for a longevity study. The findings of the present study significantly contribute to the field of aging research and provide information on the elderly healthy dietary nutrition strategy through analysis of the nutrient intake and plasma metabolites of a population derived from long-lived regions.

However, this preliminary exploratory study has some limitations. Firstly, health and longevity are modulated by several other factors, such as genes, environment, and lifestyle [38]. The effects of these factors on healthy longevity should be evaluated in further studies, and confounding factors should be excluded. Secondly, the current study used a cross-sectional design; thus the findings may not indicate a causal relationship between metabolites and longevity. Notably, the specific metabolic profile of the long-lived elderly from the LR was evaluated in the present study, providing novel insights and a basis for further research on the relationship between longevity and metabolism. Future studies should be conducted to evaluate the gut microbiota structure and metabolites in long-lived populations to further evaluate the mechanisms underlying healthy aging. Thirdly, the NMR spectroscopy technique used in the study was only based on a one-dimensional analysis of proton (1D) NMR data, thus limiting interpretations of the results on metabolites. Therefore, two-dimensional (2D) approaches, such as J-Resolved or heteronuclear multiple bond correlation spectroscopy (HMBC) and heteronuclear single quantum coherence (HSQC), should be used to explore the metabolites involved in metabolic pathways to improve the accuracy of the findings [39]. Moreover, a larger sample size, prospective studies, and use of other biofluids, such as fecal or urinary samples, should be applied to explore more biomarkers associated with health and longevity, to verify the results from metabolomic analysis, and to evaluate nutrient intake characteristics. Further, the results should be discussed and interpreted from the perspective of previous studies and of the study hypotheses.

## 5. Conclusions

To the best of our knowledge, the present study is the first one to report the associations between the nutrient intake and plasma metabolites of healthy long-lived humans. The plasma metabolites correlated with health and longevity included VLDL, lactate, alanine, NAG, citrate, tyrosine, choline, carnitine, TMAO, β-glucose, α-glucose, valine, and unsaturated lipids. Notably, these metabolites were associated with a significant enrichment of glycolysis/gluconeogenesis; aminoacyl-tRNA biosynthesis; and alanine, aspartate, and glutamate metabolism pathways. The results showed that the levels of unsaturated lipid, α-glucose, β-glucose, and TMAO were significantly correlated with the age of the participants from the longevous region. These findings provide a basis for further elucidating the mechanisms underlying healthy aging in longevity individuals. The analysis of dietary patterns and metabolomic profiles showed that decreased energy, protein, and fat intake, along with elevated dietary fiber and choline intake, are might associated with good health and longevity. These findings provide novel insights for future research on nutritional metabolomics. In addition, this research provides some reference value for the older adults’ dietary nutrition guidance and healthy recipe development. Moreover, the findings provide a basis for the further identification of potential longevity-related metabolic biomarkers for understanding the relationship between human metabolism and longevity.

## Figures and Tables

**Figure 1 nutrients-14-02539-f001:**
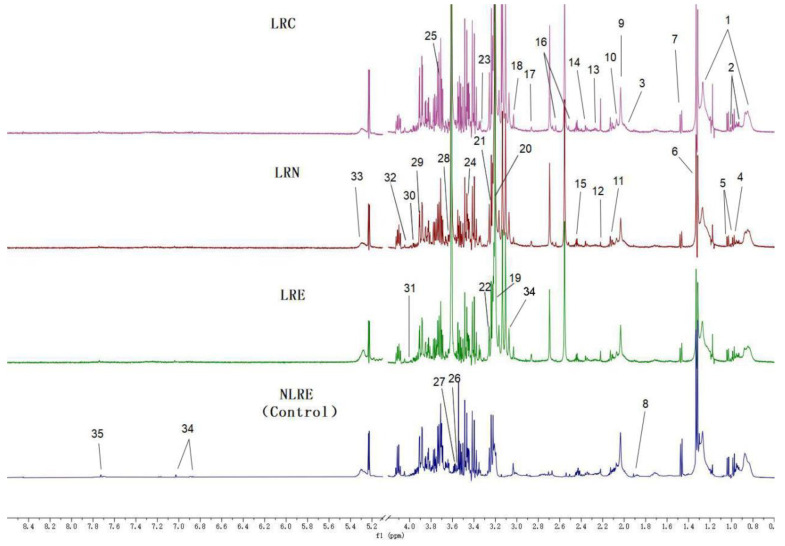
^1^H NMR spectra (500 MHz) of plasma metabolites from samples of healthy subjects at different ages from the longevous region (LRC, LRN, and LRE) and control region (NLRE). Assignments and marks of plasma metabolites: (1) lipids (mainly VLDL); (2) isoleucine; (3) proline; (4) leucine; (5) valine; (6) lactate; (7) alanine; (8) acetic acid; (9) *N*-acetyl glycoprotein (NAG); (10) glutamic acid; (11) glutamine; (12) propanone; (13) acetoacetic acid; (14) pyranic acid; (15) succinic acid; (16) citrate; (17) trimethylamine; (18) creatine; (19) choline; (20) carnitine; (21) phosphorylcholine; (22) trimethylamine oxide (TMAO); (23) scyllitol; (24) β-glucose; (25) α-glucose; (26) glycine; (27) threonine; (28) inositol; (29) glycerophosphocholine (GPC); (30) 1-methylhistidine; (31) creatine; (32) triglyceride; (33) unsaturated lipid; (34) tyrosine; (35) histidine.

**Figure 2 nutrients-14-02539-f002:**
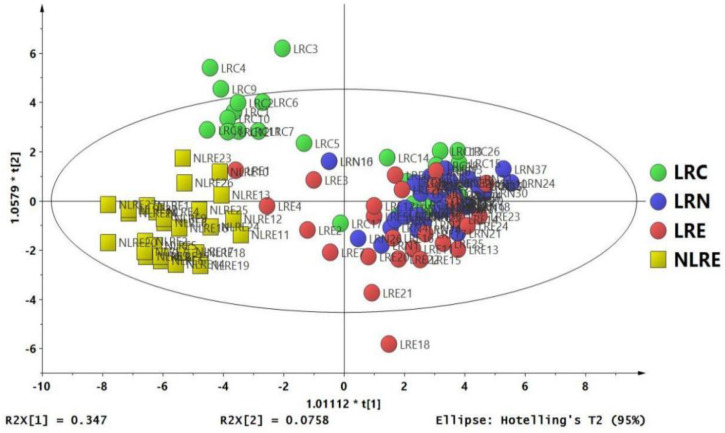
OPLS-DA score plot of the LRC, LRN, LRE, and NLRE groups showing the clustering of samples in the training set.

**Figure 3 nutrients-14-02539-f003:**
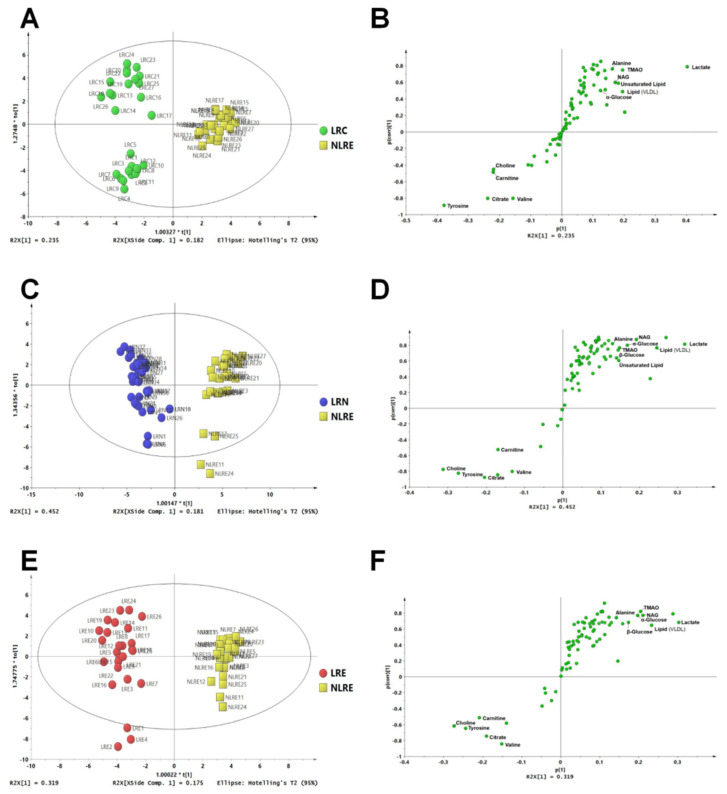
Plasma metabolic profiles of participants from the longevous region and control region. OPLS-DA score plots (**A**,**C**,**E**) showing clustering of metabolites, and s-plot (**B**,**D**,**F**) showing identified metabolites. Comparisons between LRC and NLRE (**A**,**B**), between LRN and NLRE (**C**,**D**), and between LRE and NLRE (**E**,**F**) showing metabolites associated with longevity.

**Figure 4 nutrients-14-02539-f004:**
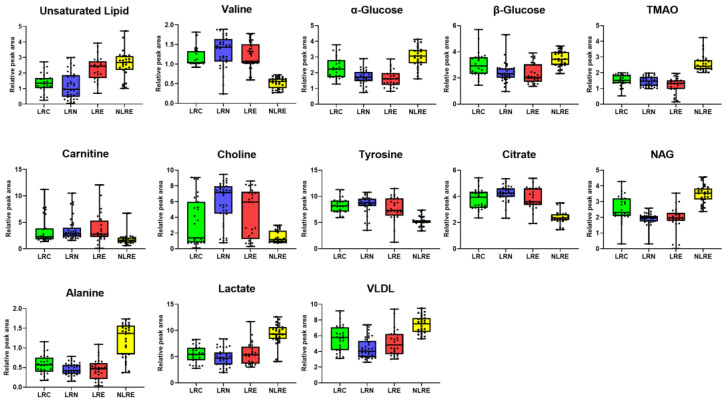
Box plot of differential metabolites levels among LRC (green), LRN (blue), LRE (red), and NLRE (yellow) groups.

**Figure 5 nutrients-14-02539-f005:**
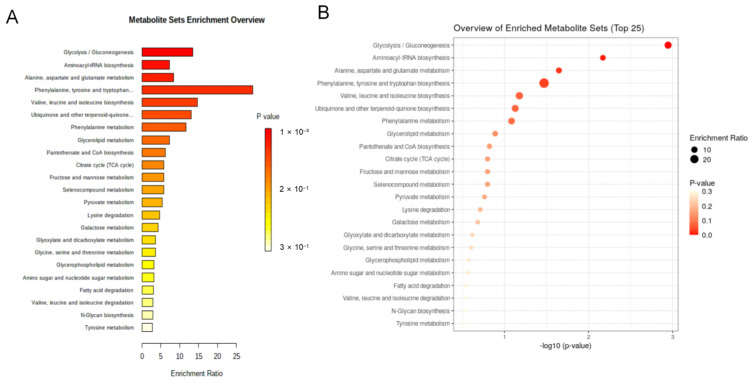
Significantly enriched metabolic pathways associated with differentially expressed pathways identified using the MetaboAnalyst 5.0 tool (4 March 2022), Metabolite sets enrichment overview (**A**); overview of enriched metabolite sets of the top 25 (**B**): (1) glycolysis/gluconeogenesis (*p* < 0.01); (2) aminoacyl-tRNA biosynthesis (*p* < 0.01); (3) alanine, aspartate, and glutamate metabolism (*p* < 0.05); (4) phenylalanine, tyrosine, and tryptophan biosynthesis (*p* < 0.05).

**Figure 6 nutrients-14-02539-f006:**
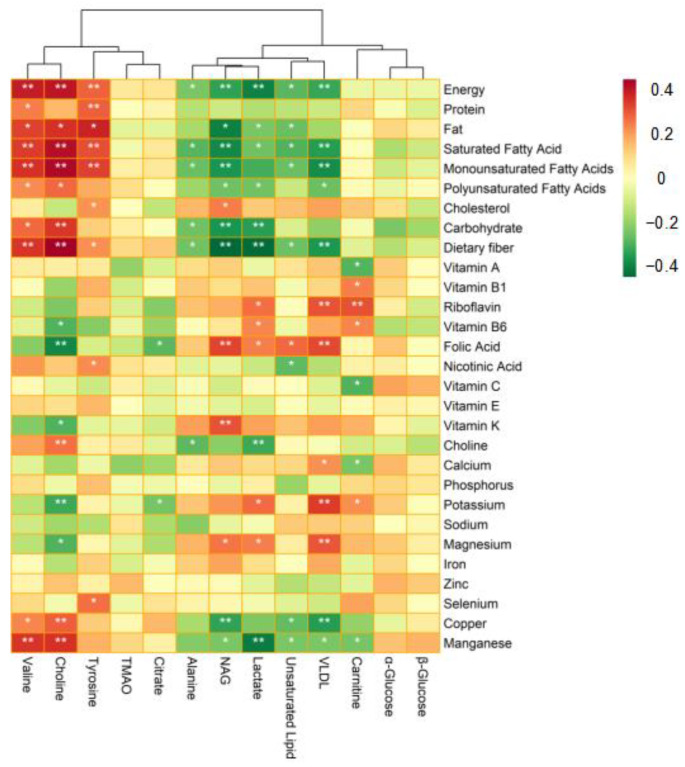
Heatmap showing the Spearman rank correlations between nutrient intake and differential metabolites. Red color indicates a positive correlation, whereas green color indicates a negative correlation; * represents *p* < 0.05; ** denotes *p* < 0.01. Analysis was conducted after adjustment for age, gender, and BMI. Correlation coefficients between dietary fiber intake and choline (r = 0.431, *p* < 0.01), NAG (r = −0.412, *p* < 0.01), and lactate (r = −0.401, *p* < 0.01) metabolites were high.

**Figure 7 nutrients-14-02539-f007:**
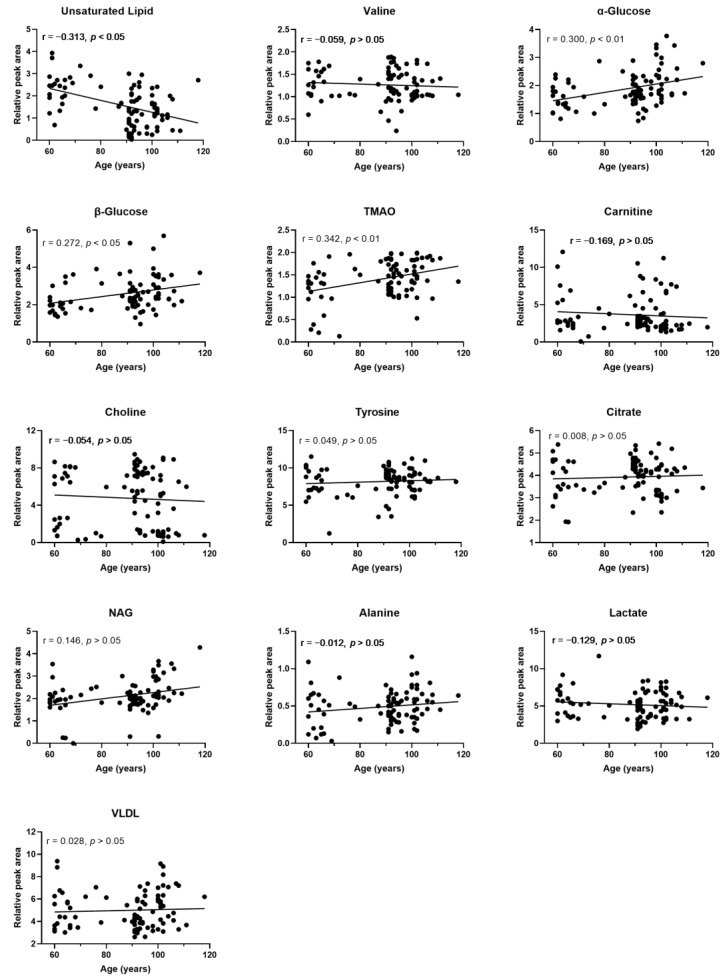
Relationship between plasma signature metabolites and age. Rank tests with Pearson’s correlation coefficient were used to evaluate correlations between plasma signature metabolites and age of longevous region participants (adjusted for gender and BMI).

**Table 1 nutrients-14-02539-t001:** Demographic and anthropometric characteristics of the participants.

	LRC (*n* = 27)	LRN (*n* = 37)	LRE (*n* = 26)	NLRE (*n* = 27)
Age	103.41 ± 4.14	93.00 ± 2.17	70.08 ± 8.24	71.84 ± 7.59
Height (cm)	142.67 ± 6.59	145.08 ± 8.44	157.65 ± 9.07	157.48 ± 5.96
Weight (Kg)	39.25 ± 4.94	40.25 ± 9.27	56.29 ± 13.97	57.71 ± 9.57
BMI (kg/m^2^)	19.30 ± 2.18	19.01 ± 3.29	22.33 ± 3.91	23.23 ± 3.3
Female/Male	3/24	8/29	9/17	14/13

Values represent means ± standard deviation (SD). LRC, centenarians from longevous regions; LRN, nonagenarians from longevous regions; LRE, elderly people aged 60–89 years from longevous regions; NLRE, elderly people aged 60–89 years from control region. BMI, Body Mass Index.

**Table 2 nutrients-14-02539-t002:** Nutrient intake levels among participants in the LRC, LRN, LRE, and NLRE groups.

	LRC	LRN	LRE	NLRE
Energy (Kcal)	1313.46 ± 107.72 ^a^	1398.75 ± 108.07 ^a^	1586.03 ± 155.5 ^b^	1742.58 ± 114.61 ^c^
Protein (g)	39.9 ± 3.65 ^a^	40.97 ± 3.24 ^a^	54.58 ± 6.22 ^b^	61.01 ± 7.22 ^c^
Fat (g)	40.05 ± 5.29 ^a^	42.39 ± 4.02 ^a^	62.81 ± 10.21 ^b^	76.73 ± 8.18 ^c^
SFA (g)	4.98 ± 0.57 ^a^	5.13 ± 0.77 ^a^	6.91 ± 1.1 ^b^	9.08 ± 1.16 ^c^
MUFA (g)	8.13 ± 1.00 ^a^	8.31 ± 0.85 ^a^	11.62 ± 1.91 ^b^	15.38 ± 1.99 ^c^
PUFA (g)	7.21 ± 0.9 ^a^	7.53 ± 1.18 ^a^	10.55 ± 1.64 ^b^	14.87 ± 2.33 ^c^
Cholesterol (mg)	189.96 ± 46.55 ^a^	197.75 ± 15.59 ^a^	311.84 ± 66.91 ^b^	378.46 ± 168.6 ^c^
Carbohydrate (g)	213.56 ± 20.75 ^b^	218.46 ± 16.86 ^b^	226.95 ± 13.78 ^b^	207.15 ± 22.47 ^a^
Dietary fiber (g)	29.26 ± 3.61 ^b^	29.93 ± 2.56 ^b^	30.73 ± 2.15 ^b^	16.93 ± 1.48 ^a^
Vitamin A (μgRE)	1287 ± 590.19 ^b^	1313.28 ± 227.57 ^b^	1592.8 ± 616.39 ^c^	855.75 ± 376.23 ^a^
Vitamin B_1_ (mg)	0.68 ± 0.06 ^a^	0.69 ± 0.04 ^a^	0.89 ± 0.10 ^ab^	0.95 ± 0.11 ^b^
Riboflavin (mg)	0.79 ± 0.10 ^a^	0.82 ± 0.07 ^a^	1.05 ± 0.15 ^b^	1.09 ± 0.17 ^b^
Vitamin B_6_ (mg)	0.49 ± 0.05 ^a^	0.49 ± 0.04 ^a^	0.52 ± 0.09 ^ab^	0.55 ± 0.25 ^a^
Folic Acid (μg)	174.83 ± 16.72 ^a,b^	169.8 ± 10.96 ^a^	187.92 ± 25.00 ^b^	225.12 ± 101.68 ^c^
Nicotinic Acid (mg)	9.5 ± 1.22 ^a^	10.42 ± 1.45 ^a^	13.84 ± 1.80 ^b^	15.72 ± 3.00 ^c^
Vitamin C (mg)	69.77 ± 17.39 ^a,b^	67.49 ± 13.7 ^a,b^	61.24 ± 19.22 ^a^	74.56 ± 21.28 ^b^
Vitamin E (mg)	19.85 ± 1.05 ^a,b^	19.57 ± 1.44 ^a,b^	17.77 ± 2.17 ^a^	22.61 ± 2.77 ^b^
Vitamin K (μg)	393.57 ± 39.25 ^a,b^	404.29 ± 32.3 ^a,b^	380.22 ± 60.55 ^a^	422.27 ± 226.75 ^b^
Choline (mg)	162.62 ± 22.09 ^b^	172.48 ± 23.2 ^b,c^	181.56 ± 23.42 ^c^	135.75 ± 21.09 ^a^
Calcium (mg)	450.19 ± 52.66 ^a,b^	452.25 ± 51.24 ^a,b^	431.91 ± 98.93 ^a^	475.04 ± 114.91 ^b^
Phosphorus (mg)	702.25 ± 54.72 ^a^	720.16 ± 69.12 ^a^	818.21 ± 101.72 ^a,b^	954.99 ± 86.54 ^b^
Potassium (mg)	1580.6 ± 158.27 ^a^	1612.34 ± 108.16 ^a^	1796.21 ± 270.81 ^a,b^	1918.65 ± 367.79 ^b^
Sodium (mg)	1648.18 ± 102.65 ^a^	1688.02 ± 252.41 ^a,b^	1801 ± 251.74 ^b^	2298 ± 274.3 ^c^
Magnesium (mg)	323.07 ± 32.81 ^a^	385.02 ± 26.40 ^b^	430.03 ± 54.7 ^b^	330.75 ± 25.02 ^a^
Iron (mg)	13.33 ± 1.74 ^a^	13.91 ± 1.69 ^a^	15.94 ± 3.59 ^b^	15.6 ± 1.95 ^b^
Zinc (mg)	6.06 ± 1.76 ^a^	6.66 ± 1.02 ^a^	7.87 ± 1.99 ^b^	8.08 ± 2.07 ^b^
Selenium (μg)	24.65 ± 3.73 ^a^	26.43 ± 2.38 ^a^	31.84 ± 5.92 ^b^	32.05 ± 7.04 ^b^
Copper (mg)	3.37 ± 0.67 ^a^	3.72 ± 0.72 ^b^	3.75 ± 0.64 ^b^	3.24 ± 0.88 ^a^
Manganese (mg)	3.05 ± 0.67 ^a^	3.22 ± 0.63 ^a^	3.88 ± 0.58 ^b^	3.65 ± 0.61 ^b^

Values are presented as means ± SD. Values with different superscript letters across a row are significantly different, *p* < 0.05. LRC, centenarians from longevous regions; LRN, nonagenarians from longevous regions; LRE, elderly people aged 60–89 years from longevous regions; NLRE, elderly people aged 60–89 years from control region. SFA, saturated fatty acid; MUFA, monounsaturated fatty acids; PUFA, polyunsaturated fatty acids.

**Table 3 nutrients-14-02539-t003:** Characteristic plasma metabolites associated with longevity.

	Chemical Shift (ppm)	FC ^1^		
		LRC vs. NLRE	LRN vs. NLRE	LRE vs. NLRE
Lipid (VLDL)	0.86	−0.37	−0.77	−0.69
Lactate	1.32	−0.79	−0.97	−0.74
Alanine	1.47	−1.45	−1.41	−1.44
NAG	2.03	−0.45	−0.87	−0.86
Citrate	2.53	+0.66	+0.78	+0.64
Tyrosine	3.14	+0.68	+0.70	+0.55
Choline	3.20	+0.96	+2.18	+1.79
Carnitine	3.21	+0.62	+1.20	+1.27
TMAO	3.26	−0.75	−0.83	−1.10
β-Glucose	3.46	/	−0.49	−0.61
α-Glucose	3.53	−0.38	−0.78	−0.88
Valine	3.60	+1.20	+1.36	+1.30
Unsaturated Lipid	5.29	−0.91	−1.09	/

^1^ FC (fold change) denotes the fold change of metabolites between two groups; that is, LRC/NLRE, LRC/NLRE, and LRE/NLRE, which are the logarithmic values (base 2) of the mean ratio between the two groups; + indicates the latter is relatively higher, and − indicates that the latter is relatively lower.

## Data Availability

The data in this study are available on request from the corresponding author.

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
