# Peer review of "1H NMR-Based Metabolomics Reveals the Intrinsic Interaction of Age, Plasma Signature Metabolites, and Nutrient Intake in the Longevity Population in Guangxi, China"

_nutrients, 2022, doi:10.3390/nu14122539_

Round 1
Reviewer 1 Report
I’ve read with attention the paper of Li et al. that is potentially of interest. The background and aim of the study have been clearly defined. The methodology applied is overall correct, the results are reliable and adequately discussed. I’ve only some minor comments:
- Acronyms should be also spelled below the tables, in order to improve the reading
- Comparative analyses should be adjusted for multiple comparisons, while the Spearmen correlation test is also not adjusted and consequently only orientative. Is it not possible to carry out any kind multiple regression analysis?
- A relevant limitation that should be stressed (and only indirectly mentioned by the authors) is that the control group was obtaine in a region that had a priori a lower rate of centenarians and that could be related by genetic as well as environmental factors totally independent from blood nutrient levels.
Author Response
Dear reviewer,
Thank you for your review and guidance.
Please see the attachment.

Reviewer 2 Report
In this paper, the investigators report 1H NMR metabolomics findings on serum metabolite profiles they have collected from long-lived individuals and present a correlation analysis indicating a relationship between age, plasma metabolite levels, and nutrient intake.
This study appears to be the first one to date to report the association between nutrient intake and plasma metabolite levels in long-lived humans.
Strength: Focus of the study is interesting; investigators have access to a unique cohort of long-lived individuals and the question of what are mechanisms that enable some individuals to live a healthy long life as compared to healthy individuals with a shorter lifespan is interesting and worthy of further studies.
While the manuscript has merit, there are several issues with the manuscript that should be addressed. Several items relate to the specific words employed and the potentially misleading impression this could give to a reader. There are also major issues with the accuracy of the metabolite annotation (it seems that none of the metabolite IDs were validated using either 2D NMR methods or spiking of standards) and the interpretation of the results, regarding what the differences in metabolite level may mean from a physiological point of view.
There is also great concern about relying on only 13 metabolites to conduct metabolic pathway impact analysis in MetaboAnalyst. The reviewer is skeptical that all the pathways that the authors reported as being differentially impacted in the LR versus the NLR groups are in fact real. Profiling of a lot more metabolites would need to be undertaken before a detailed metabolic pathway analysis can be undertaken.
Below are included specific requests for revisions before the manuscript can be further evaluated as to whether or not it is acceptable for publication in Nutrients.
Revisions requested:
First in the abstract, lines 14 and 15, the terms “longevous” and “non-longevous” are awkward and should be explain, so that the reader has a sense of what is meant there.
Lines 38-39, the sentence “Furthermore, the dietary components modulate metabolism in human beings and results in metabolite variations” is very vague and rather uninformative. Consider revising or deleting if the content cannot be improved.
Some of the English wording is awkward and revised
For example, instead of “The metabolite profiles are explored…” I would suggest revising as “Metabolite profiles are determined …”
Line 45 “mass spectroscopy” should be replaced by “mass spectrometry”
Line 46 “…which exhibit changes..” should be changed to something along the lines “…which reflect changes”
Line 49: the sentence starting with “This technique…”; which technique are the authors referring to? If it is 1H NMR spectroscopy, this should be stated explicitly
Line 52: “urine fluid”, should be replaced with “urine” (the word urine fluid is weird)
Lines 53-57, the authors list amino acids and lipids as being associated with human health and the process of aging. I sincerely doubt that these are the only metabolites associated with aging. I would encourage the authors to revise their statement.
Similarly, lines 57-58: the authors state “Populations that live long especially centenarians exhibit healthy and good aging patterns.” While this seems to make intuitive sense, I doubt that this is true and universal. I recommend being a little less absolute when making such broad statements.
Lines 68-69, the sentence starting with “Combination of metabolomic with multivariate statistical analysis …” is awkward and the authors are encouraged to revise.
Line 82, instead of “The findings of the current study will provide...,” I would recommend being a little less categorical and replace the word “will” with “may” or “has the potential to”
On line 90, what do the decimal points on the numbers represent? It seems that incidence would be a measure of the number of individuals that are centenarians in a population of 100,000 citizens – so what are the fractions representing?
Lines 100-101, I am concerned about the impact of having 90 people in the LR group and only 27 people in the NLR group. The number of participants in the two different groups is very unbalanced, and I worry that this skewed distribution will interfere with the multivariate statistical analyses and the selection of which metabolites are most important for distinguishing the LR group from the NLR group. It would be helpful if the authors could address this issue and and clarify the potential impact of an uneven distribution of participants in the two in the text.
In the NMR data acquisition and analysis section, line 141, “number of sampling sites” should be replaced with “with number of data points” (which are used for digitization of the FIDs (free induction decays).
Line 143, “Spectra data were baseline corrected…” should be changed to “1H NMR spectra were baseline corrected….” and which baseline correction function was used?
Line 153, the sentence structure needs some correction: the word “Plasma” should be lower case, and the phrase “..by comparing the spectrum collected from NMR with the chemical shifts and peaks of standard compounds..” should be replaced by clearer wording such as, for example, “ …..by comparing the NMR spectra with the chemical shifts and 1H spectral patterns of reference compounds from the Chenomx spectral database of small molecule metabolites for 500 MHz (11.7 Tesla) magnetic field strength NMR…”
I was also curious how the authors integrated binning of the spectra with the metabolite identification approach of chenomx? I am not sure I understand how the spectral binning helped identify and quantify metabolites when using chenomx.
On line 167, it is said that permutation tests were conducted to assess the validity of the supervised PLS-DA and OPLS-DA model – however, n is stated to be equal to 200 – which seems too few permutations; How do the validation metrics measure to when an n or 1,000 or 2,000 is used?
On line 169, the authors state that metabolites were considered significant if their VIP scores were greater than 1. This seems to be a low bar; we normally consider metabolites to be significant when VIP scores are greater than 1.2 or 1.4. What happens to the results when only the metabolites with VIP > 1.2 are selected? I would expect that they would be fewer significant metabolites, and this could be important, as only 13 metabolites were identified and quantified in this study. It would be good if the authors clarify this issue and provide a stronger rationale for their selection of VIP thresholds.
Lastly, for the materials and methods section, the description of the spearman correlation analysis (section on lines 176-181) is a bit brief. It would be helpful to provide more details on how the correlation analyses were conducted to investigate which metabolite level changes correlated with nutrient intake and how this was adjusted for age, gender, and BMI
On line 177, the authors introduce the term “differentially expressed metabolites” and use this term throughout the text. This is in fact inaccurate and should be reworded throughout the text. Genes are differentially expressed; metabolites are produced by proteins, and what is observed are differences in levels or concentrations of metabolites. Please make sure that the correct terminology is used throughout the text.
Author Response

(The authors gave the same response as above.)

Reviewer 3 Report
The current study tries to show differences in plasma metabolites and their dietary patterns over the length of life in an area of ​​China.
I don't really understand the geographical distribution of the participants. The authors name three locations (Donglan, Fengshan, and Dahua counties) but then speak of four zones according to population longevity.
In table 1 you should explain the meaning of the acronyms. It's not very clear in the text.
-LRC group (age: 100-118 years
-LRN group (age: 90-99 years),
-LRE group (age: 60- 89 years),
-NLRE group ?
The small number of patients included in the study is striking. I understand that it is difficult to find elderly people over 100 years old, but between 90 and 99 years old it is easier and between 60 and 89 years old it is very easy. Regarding the number contributed in each group, I do not see that the sample size has been calculated. With such a small number of participants, the representativeness of the sample is scarce and the results are not very useful for the general population.
The average daily nutritional intake is from current intake and longevity I think is more related to intake from many previous years. So I don't think it's relevant. It is well known that the older the individual, the less calories and less protein they ingest, possibly due to chewing problems.
On the other hand, Pearson correlation analysis shows a relationship between plasma signature metabolites and age with a very low intensity, which makes its significance less interesting. Therefore, the conclusion drawn by the authors is not exact. “The results showed that the levels of unsaturated lipid, α - glucose, β-glucose, and TMAO were significantly correlated with the age of participants from the longevous region ALTHOUGH WITH A VERY LOW INTENSITY”.
With all these limitations, the statements made by the authors are very risky because nothing can be guaranteed with these data. For example: “These results indicate that lower levels of lipid metabolism is a key characteristic of long-lived individuals”
In addition, it is a cross-sectional study and therefore, the results cannot be translated in terms of causality. You cannot say: “the findings of the present study revealed that α -glucose, β -glucose, and TMAO levels increased with age….”
I would not dare to say that "decreased energy, protein and fat intake, and elevated dietary fiber and choline intake are associated with good health and longevity" because it is a cross-sectional study and causality cannot be established. What has really been found is that older people eat less energy, protein and fat.
Author Response

(The authors gave the same response as above.)

Reviewer 4 Report
In the present study, diet-related signature metabolites of longevity were evaluated by analyzing plasma metabolites and dietary characteristics of the longevity population in the Hechi region. The current study's findings will provide a theoretical basis for establishing healthy dietary plans, exploring mechanisms underlying the aging process and designing strategies for achieving healthy aging. Although the overall interest and visibility of this work, some aspects should still be considered to improve the quality and objectiveness.
1) The background of the study should be made very clear. Provide more details of the introduction and review of the work.
2) Please speculate about the reasons for the obtained results. The discussion needs to improve.
3) In Conclusion, the authors should add the potential practical application.
4) The article should be reviewed for English language proficiency and grammar. There are a lot of sentences without sense, misspelled words, and punctuation errors.
Author Response

(The authors gave the same response as above.)

Round 2
Reviewer 3 Report
The article has gained in clarity. I still think that there are few patients in the control group and that the sample size has not been calculated.Reviewer 4 Report
Requested corrections were completed.